

# Phosphorylation of the N-terminus of Syntaxin-16 controls interaction with mVps45 and GLUT4 trafficking in adipocytes

Shaun K. Bremner[1], Rebecca Berends[2], Alexandra Kaupisch[2], Jennifer Roccisana[2], Calum Sutherland[3], Nia J. Bryant[4] and Gwyn W. Gould[1]

[1] SIPBS, University of Strathclyde, Glasgow, United Kingdom
[2] Institute of Molecular, Cell and Systems Biology, University of Glasgow, Glasgow, United Kingdom
[3] Department of Cellular Medicine, University of Dundee, Dundee, United Kingdom
[4] Department of Biology, University of York, York, United Kingdom

Corresponding author
Shaun K. Bremner,
shaun.bremner@strath.ac.uk

## ABSTRACT

The ability of insulin to stimulate glucose transport in muscle and fat cells is mediated by the regulated delivery of intracellular vesicles containing glucose transporter-4 (GLUT4) to the plasma membrane, a process known to be defective in disease such as Type 2 diabetes. In the absence of insulin, GLUT4 is sequestered in tubules and vesicles within the cytosol, collectively known as the GLUT4 storage compartment. A subset of these vesicles, known as the 'insulin responsive vesicles' are selectively delivered to the cell surface in response to insulin. We have previously identified Syntaxin16 (Sx16) and its cognate Sec1/Munc18 protein family member mVps45 as key regulatory proteins involved in the delivery of GLUT4 into insulin responsive vesicles. Here we show that mutation of a key residue within the Sx16 N-terminus involved in mVps45 binding, and the mutation of the Sx16 binding site in mVps45 both perturb GLUT4 sorting, consistent with an important role of the interaction of these two proteins in GLUT4 trafficking. We identify Threonine-7 (T7) as a site of phosphorylation of Sx16 *in vitro*. Mutation of T7 to D impairs Sx16 binding to mVps45 *in vitro* and overexpression of T7D significantly impaired insulin-stimulated glucose transport in adipocytes. We show that both AMP-activated protein kinase (AMPK) and its relative SIK2 phosphorylate this site. Our data suggest that Sx16 T7 is a potentially important regulatory site for GLUT4 trafficking in adipocytes.

## INTRODUCTION

The regulation of glucose transport is a central facet of insulin action (*Saltiel, 2021*; *Santoro, McGraw & Kahn, 2021*). Upon insulin binding its receptor, intracellular GLUT4-containing vesicles are rapidly mobilised to the cell surface where they dock and fuse, resulting in increased levels of GLUT4 in the plasma membrane and a consequent increase in glucose transport (*Gould, Brodsky & Bryant, 2020*; *Klip, McGraw & James, 2019*; *Santoro, McGraw & Kahn, 2021*).
Key to this effect of insulin is the ability of fat and muscle to sequester GLUT4 in tubules and vesicles throughout the cytoplasm which are collectively referred to as the GLUT4 storage compartment (*Bogan & Kandror, 2010*; *Gould, Brodsky & Bryant, 2020*). A subfraction of the GSC, generally referred to as insulin-sensitive intracellular vesicles (IRVs), is selectively mobilized by insulin to the plasma membrane (*Bogan & Kandror, 2010*; *Klip, McGraw & James, 2019*). The trafficking itinerary of GLUT4 in insulin-sensitive cells has been the subject of significant research. In the absence of insulin GLUT4 is sequestered intracellularly by the coordinated action of two intracellular trafficking pathways (*Bryant, Govers & James, 2002*; *Klip, McGraw & James, 2019*). The first operates between the plasma membrane and recycling endosomes and serves to efficiently internalize GLUT4. GLUT4 is then sorted from this pathway into a second cycle that operates between recycling endosomes, the *trans* Golgi network (TGN) and IRVs. Many of the molecules involved in these sorting pathways have been defined, but their regulation is less well understood (*Gould, Brodsky & Bryant, 2020*; *Klip, McGraw & James, 2019*).

Patients with Type two Diabetes Mellitus (T2DM) exhibit impaired insulin-stimulated glucose transport in muscle and fat; many studies have supported the hypothesis that GLUT4 sorting and/or trafficking are impaired in these patients (*Bogan & Kandror, 2010*; *Garvey & Birnbaum, 1993*; *Garvey et al., 1998*; *Garvey et al., 1993*; *Hunter & Garvey, 1998*; *Livingstone et al., 2022*). For example, GLUT4 is mis-sorted into a denser membrane fraction in muscles from patients with T2DM compared to controls, and insulin does not promote mobilization of GLUT4 from this fraction to the PM (*Garvey et al., 1998*). Consistent with this, recent work has revealed reduced expression of key proteins involved in GLUT4 sorting in muscle of obese diabetic patients compared to weight-matched controls (*Livingstone et al., 2022*). Similarly, defective GLUT4 sorting in adipocytes isolated from patients with T2DM (*Garvey & Birnbaum, 1993*) and with gestational diabetes (*Garvey & Birnbaum, 1993*) has been described. Consequently, there is an imperative to understand how the trafficking of GLUT4 into IRVs is regulated.

Communication between intracellular membrane compartments by vesicle trafficking uses components of the SNARE machinery (*Sudhof & Rothman, 2009*). Studies from several groups, including our own, have identified Syntaxin 16 (Sx16) as a key $Q_a$-SNARE involved in GLUT4 trafficking towards IRVs (*Bremner et al., 2022*; *Perera et al., 2003*; *Proctor et al., 2006*; *Shewan et al., 2003*). Knockdown or knock-out of Sx16 impairs insulin-stimulated glucose transport and results in a reduction in GLUT4 levels (*Bremner et al., 2022*; *Proctor et al., 2006*). $Q_a$-SNAREs are regulated by interaction with members of the Sec1/Munc18 (SM) family (*Sudhof & Rothman, 2009*); Sx16 binds the SM protein mVps45 (*Tellam et al., 1997*), a protein also required for sorting of GLUT4 into IRVs (*Roccisana et al., 2013*). This interaction is complex however, and likely involves multiple 'modes' of interaction between the $Q_a$ SNARE and the SM protein (*Munson & Bryant, 2009*). In the case of the yeast homologues of Sx16 and mVps45 (Tlp2p and Vps45p, respectively) the interaction of Tlg2p with Vps45 involves a high affinity interaction between the N-terminus of the $Q_a$ SNARE with a hydrophobic 'pocket' on the surface of Vps45 which is essential for the formation of productive SNARE complexes (*Carpp et al., 2006*; *Furgason et al., 2009*; *Struthers et al., 2009*).
Previously published data suggested that Sx16 and mVps45 act together in GLUT4 sorting (*Bremner et al., 2022*; *Proctor et al., 2006*; *Roccisana et al., 2013*). How the interaction between these proteins is regulated, and contributes to their function, remains to be determined, but it is of note that we have previously shown that Sx16 is a phosphoprotein (*Perera et al., 2003*), and that levels of Sx16 are significantly reduced in obese patients with T2DM compared to obese, non-diabetic individuals (*Livingstone et al., 2022*).

Here we show that interaction between Sx16 and mVps45 is required for the effective sorting of GLUT4 in adipocytes and that this interaction may be regulated by phosphorylation of Sx16 in the N-terminal regulatory domain. We identify Threonine-7 (T7) as a site of phosphorylation of Sx16 *in vitro* and show over-expression of Sx16 containing the phosphomimetic T7D mutation in 3T3-L1 adipocytes significantly impaired insulin-stimulated glucose transport. These and other data argue that the N-terminus of Sx16 is a potential regulatory site for the control of GLUT4 sorting in adipocytes. Finally, we show that both AMP-activated protein kinase (AMPK) and its relative SIK2 phosphorylate this site suggesting that T7 is a potentially important regulatory site for GLUT4 trafficking in adipocytes.

## MATERIALS AND METHODS

### Cells and antibodies

HEK293T cells (Cat# CRL-3216 RRID:CVCL_0063) and 3T3-L1 fibroblasts (Cat# CCL-92.1, RRID:CVCL_0123) were obtained from the American Tissue Culture Collection and grown and differentiated into adipocytes as outlined in (*Sadler et al., 2019*). HeLa cells expressing HA-GLUT4-GFP were as described previously (*Morris et al., 2020*). AMPK $\alpha$1/ $\alpha$2 knockout mouse embryonic fibroblasts (MEFs) and control MEFs were cultured as described previously (*Laderoute et al., 2006*). Antibodies used: anti-GLUT4 (Thermo Fisher Scientific Cat# PA1-1065, RRID:AB_2191454), anti-Sx16 (Synaptic Systems Cat# 110 162, RRID:AB_887799), anti-mVps45 (Novus Cat# NB100-2431, RRID:AB_2272935), anti-VAMP4 (Synaptic Systems Cat# 136 002, RRID:AB_887816), anti-SNAP23 (Synaptic Systems Cat# 111 202, RRID:AB_887788) and anti-Syntaxin 4 (Synaptic Systems Cat# 110 042, RRID:AB_887853). A phospho-specific antibody raised in rabbits against the N-terminal 16 amino acids of murine Sx16 with Thr-7 chemically phosphorylated and recognizing Thr-7 in the Sx16 N-terminus only when phosphorylated, was generated by Antagene Inc (Mountain View, CA, USA) and purified using affinity chromatography. Anti-AKT (pan) (Cell Signaling Technology Cat# 2920, RRID:AB_1147620) and anti-AKT Phospho-S473 (Cell Signaling Technology Cat#4058), RRID:AB_331168) were used as outlined (*Bremner et al., 2022*).

### Plasmids and viruses

HA-GLUT4-GFP in the pRRL-PGK plasmid was from Cynthia Mastick (*Brewer et al., 2011*; *Muretta & Mastick, 2009*) and pCOG70, driving the over-expression of Vps45p harboring an HA tag at its carboxy terminus was a generous gift of Dr. Lindsay Carpp (*Carpp et al., 2006*; *Carpp et al., 2007*). The V107R (Sx16 binding deficient) mutant of mVps45 (corresponding to V107R) was generated using a QuickChange kit (Thermo
Fisher Milton Park, UK) and fully sequenced on both strands prior to expression in yeast. T7A and T7D mutants in a truncated Sx16 lacking the transmembrane domain were generated by PCR and fully sequenced. Constructs comprising the first 33 residues of Sx16 (either wild-type or containing the F10A mutation previously shown to disrupt binding to mVps45) fused to Protein-A were generated by PCR and fully sequenced on both strands and sub-cloned into pcDNA3.1 for expression in mammalian cells. All other cDNAs were as previously outlined (*Morris et al., 2020*; *Proctor et al., 2006*; *Sadler et al., 2019*). Full length Sx16 containing the T7A to T7D mutations were synthesised by GenScript (Piscataway, NJ, USA) and sub-cloned into pCDH vectors for lentivirus production using the ViraPower™ Lentiviral system (Invitrogen, Waltham, MA, USA) following manufacturer's instructions. The sequence used corresponds to Q8BVI5 in UniProt and is the largest of the proposed splice variants of Sx16 in mouse tissues, and is the only version thought to contain a transmembrane domain. 3T3-L1 adipocytes were incubated with lentivirus for 12 h on day 5 or 6 post-differentiation. The media was changed, and the cells were assayed between day 7 and 9 after differentiation (48–72 h post infection).

## Recombinant protein production

Sx16 lacking its transmembrane domain, or the T7A or T7D mutants, were sub-cloned in-frame with dual protein-A tags, expressed in *E. coli* and purified on IgG-Sepharose (GE Healthcare). Protein levels were determined using Micro-BCA kits (Pierce, Waltham, MA, USA).

## 2-deoxyglucose transport, adipocyte fractionation, immunofluorescence, and immunoblotting

2-Deoxy-D-glucose transport was assayed in cells exactly as outlined (*Bremner et al., 2022*; *Roccisana et al., 2013*) in except 96-well plates were employed to minimise virus requirements. To determine the levels of GLUT4, adipocyte lysates were generated as outlined (*Roccisana et al., 2013*; *Sadler et al., 2019*). SDS-PAGE and immunoblotting were performed as described (*Sadler et al., 2019*). Immunoblot signals were quantified using LICOR software (*Bremner et al., 2022*). Immunofluorescence to quantify cell surface GLUT4 staining in HeLa cells was performed exactly as outlined in *Morris et al. (2020)*. Confocal microscopy of HA-GLUT4-GFP-expressing cells was performed on a Zeiss Exciter equipped with a 63x lens (*Morris et al., 2020*). Images were captured and processed using Zeiss software.

## Phosphorylation and analysis

Adipocyte cytosol was prepared from day-10 adipocytes. Cell plates were washed three times in ice-cold buffer with protease inhibitors (50 mM Hepes pH 7.4; 5 mM EDTA, 10 mM sodium pyrophosphate, 10 mM NaF, 150 mM NaCl, 2 mM sodium orthovanadate, 50 mM beta-glycerophosphate and 1 mM DTT) and homogenised by passing through a needle twenty times. The homogenate was centrifuged at 12,000 $g$ for ten minutes and the fat cake carefully aspirated. The supernatant was used in a series of *in vitro* kinase assays; supernatant was stored at $-80\,^{\circ}$C prior to use. In brief, 30 µg of cytosol extract was incubated in 50mM HEPES pH 7.5, 10 mM $MgCl_2$, 2 mM $MnCl_2$, 0.2 mM DTT, 50µM

ATP.Mg$^{2+}$ ($\pm[\gamma$-$^{32}$P]-ATP) in the presence of 5 µg of recombinant Protein A-tagged Sx16 (residues 1-279 of Sx16, fused to two copies of Protein-A *via* a thrombin cleavage site), protein A alone or myelin basic protein at 30 °C for 2 h. Samples were analysed by autoradiography.

Analysis of the phosphorylation sites was performed at the FingerPrints facility, University of Dundee. Phosphorylation reactions were carried out exactly as outlined above but using non-radioactive ATP; control reactions lacking cytosol and/or ATP were performed. Bands corresponding to Sx16-PrA were excised and trypsinised, followed by liquid chromatography-electrospray ionisation mass spectrometry on an ABI QTrap 4000 instrument. Potential phospho-peptide ions present in samples but absent from controls were submitted to ESI-MS/MS to determine their amino acid sequence and locate the phosphorylated residue.

A total of 2 µg of purified ProteinA-Sx16 or Protein A-Sx16-T7A was incubated with the indicated purified recombinant protein kinases (5 mU; see figure legends for details) in the presence of 10 mM MgCl$_2$, 0.1 mM [$\gamma$-$^{32}$P]-ATP (approx. 0.5 × 10$^6$ cpm/nmol) and kinase buffer (50 mM Tris-Cl, 0.03% (v/v) Brij-35, 0.1% (v/v) $\beta$-mercaptoethanol) at 30 °C in a total volume of 50 µl. A 20 µl aliquot of reaction mix was taken at the times indicated on the figure legends and the reactions terminated by addition of 4xNovex SDS PAGE loading buffer and heating at 70 °C for 15 mins. The Sx16 was separated from [ $\gamma$-$^{32}$P]-ATP by SDS-PAGE on Novex 4–12% gradient gels using a MOPS buffer. After staining with Coomassie Brilliant Blue to visualise the protein bands, the gels were dried and subjected to autoradiography. Finally, the Sx16 bands were excised, and the incorporated radioactivity was determined by scintillation counting.

In experiments using anti-phospho-T7 for detection, *in vitro* kinases assays typically contained 0.5 mg of bacterially expressed Sx16 incubated with 30 µg of cytosol in the presence or absence of 25 mM ATP for 60 min at 30 °C as indicated. Samples were then immunoblotted with anti-Sx16 antibodies or anti-phospho-T7 antibodies as described in the figure legends.

### Statistical analysis

Result are expressed as $\pm$ SD with the N-number indicated. Significant differences were determined using either $t$-test or ANOVA as indicated in the figure legends, with $p < 0.05$ being deemed as significant. Statistical analysis was carried out using GraphPad Prism software.

## RESULTS

### Interaction of mVps45 and Sx16 is functionally important for GLUT4 sorting

Syntaxin proteins are subject to regulation by cognate SM proteins through multiple distinct but evolutionarily conserved binding mechanisms (*Morey et al., 2017*; *Munson & Bryant, 2009*; *Sudhof & Rothman, 2009*; *Yu et al., 2013*). One of the best characterized examples of this is binding of yeast Vps45p by the Syntaxin Tlg2 *via* its N-terminal peptide which is necessary for productive SNARE complex formation (*Carpp et al., 2006*; *Carpp et*

*al., 2007*; *Furgason et al., 2009*; *Struthers et al., 2009*). This binding mechanism is conserved for mammalian Sx16 which binds mVps45 in an analogous manner (*Struthers et al., 2009*) and involves the insertion of the Sx16 N-terminus into a hydrophobic pocket located on the surface or mVps45 (*Carpp et al., 2006*; *Dulubova et al., 2002*; *Munson & Bryant, 2009*). This interaction is known to mask a lower affinity but functionally important interaction which does not involve the Sx16 N-terminus and involves Sx16 binding to mVps45 in a conformation incompatible with productive SNARE complex formation (*Furgason et al., 2009*; *Munson & Bryant, 2009*; *Struthers et al., 2009*).

Given the previously demonstrated a role for both Sx16 and mVps45 in GLUT4 traffic (*Bremner et al., 2022*; *Proctor et al., 2006*; *Roccisana et al., 2013*; *Shewan et al., 2003*), here we asked whether the interaction between these two proteins is required for their function(s) in this process. To this end, we undertook two different approaches to disrupt this interaction. We produced a protein fusion between the N-terminal 33 residues of Sx16 and Protein-A and expressed this in HeLa cells expressing HA-GLUT4-GFP. We reasoned that introduction of this N-terminal peptide into cells would inhibit the interaction between Sx16 and mVps45, prevent functional SNARE complex formation and perturb GLUT4 trafficking to the IRVs (*Munson & Bryant, 2009*) and elected to study this initially using the readily tractable HeLa cell system which we and others have shown behaves similarly to adipocytes and muscle (*Bryant & Gould, 2020*; *Morris et al., 2020*). Consistent with this, we observed significantly reduced insulin-stimulated translocation of HA-GLUT4-GFP to the plasma membrane in cells staining positive for the Protein-A fusion. By contrast, a mutation in this peptide (F10A), previously shown to reduce binding of Sx16/Tlg2p to mVps45/Vps45p (*Dulubova et al., 2002*; *Struthers et al., 2009*), was without effect (Figs. 1A, 1B).

Studies in yeast have shown that mutation of a conserved hydrophobic residue predicted to be on the outer surface of Vps45 is required for interaction with the N-peptide of Tlg2 and Sx16, with its mutation abrogating interaction (*Carpp et al., 2006*; *Dulubova et al., 2002*; *Struthers et al., 2009*). We therefore created this mutation in mVps45 and determined its effect on GLUT4 distribution. The rationale for this approach being that if cycles of association/dissociation of mVps45/Sx16 are needed for effective GLUT4 sorting, the over-expression of a mutant defective in this interaction should manifest itself by impairment of GLUT4 sorting. Transient over-expression of (V107R)-mVps45 in HeLa cells increased cell surface GLUT4 levels measured in the absence of insulin, detected using anti-HA antibodies in non-permeabilised cells, consistent with disruption of the Sx16/mVps45 interaction perturbing trafficking of GLUT4. Over-expression of wild type mVps45 was without effect. This data is shown in Fig. 2 in which cells over-expressing wild-type HA-mVps45 or (V107R)-mVps45 were identified by immunostaining with anti mVps45 (we are unable to use anti-HA to quantify mVps45 expression because of the presence of HA-tagged GLUT4 in these cells). This approach stains both endogenous and over-expressed proteins, hence in the visual analysis shown, cells expressing higher than average mVps45 staining were assigned to be positive for the transfected DNA (Fig. 2A); similar levels of the mVps45 species were expressed as judged by immunoblot analysis (Fig. 2B). To mitigate against bias in this subjective analysis, we also measured the cell
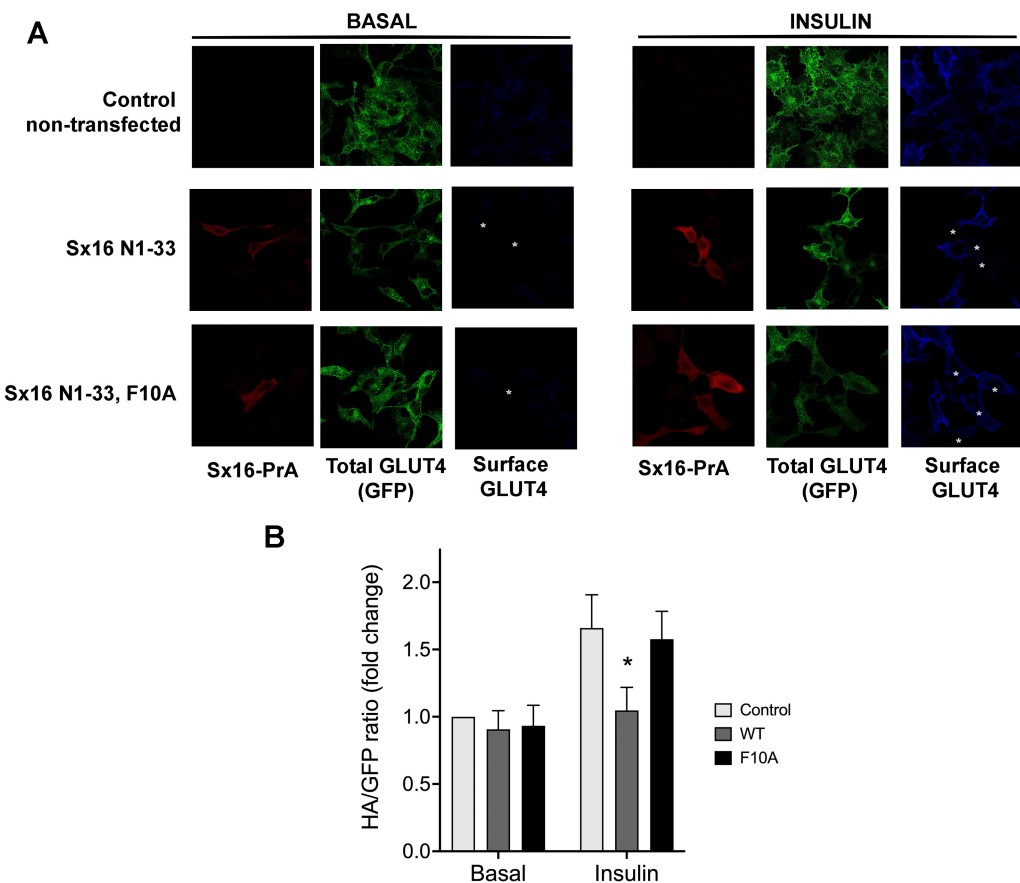

**Figure 1** **Disruption of the Sx16 N-terminus perturbs GLUT4 trafficking.** HeLa cells expressing HA-GLUT4-GFP were transfected with plasmids encoding the N-terminal 33 amino acids of Sx16 fused to protein A (WT) or the same peptide with an F10A mutation, or empty plasmid (control). 48 h after transfection, cells were stimulated with or without 1 μM insulin for 60 min and fixed. Cell surface GLUT4 was immuno-stained using anti-HA (pseudo-coloured blue) in non-permeabilised cells. After washing, cells were permeabilised and cells expressing Protein-A fusions identified using a distinct secondary antibody (pseudo-coloured red). Signal from GFP is pseudo-coloured green. (A) Data from a typical experiment; protein-A positive cells are indicated with a white asterisk in some panels so that transfected cells can be readily identified. (B) >50 cells in each condition from four separate biological replicates were quantified to allow calculation of the HA/GFP ratio; values are expressed relative to Basal control (non-transfected) cells. An asterisk (*) indicates a significant reduction in insulin-stimulated surface/total ratio compared to control and F10A insulin-stimulated cells ($p = 0.02$ ANOVA).

surface GLUT [HA staining in non-permeabilised cells]/GFP ratio for entire populations of transfected cells (*i.e.,* containing both non-transfected and transfected cells) and found that over-expression of wild-type HA-mVps45 did not change the cell surface GLUT[HA staining in non-permeabilised cells]/GFP ratio compared to non-transfected cells, but over-expression of (V107R)-mVps45 consistently increased this ratio by 20% (Fig. 2C). Collectively, these data, together with our previous observations of depletion of either protein separately (*Bremner et al., 2022*; *Proctor et al., 2006*; *Shewan et al., 2003*), supports

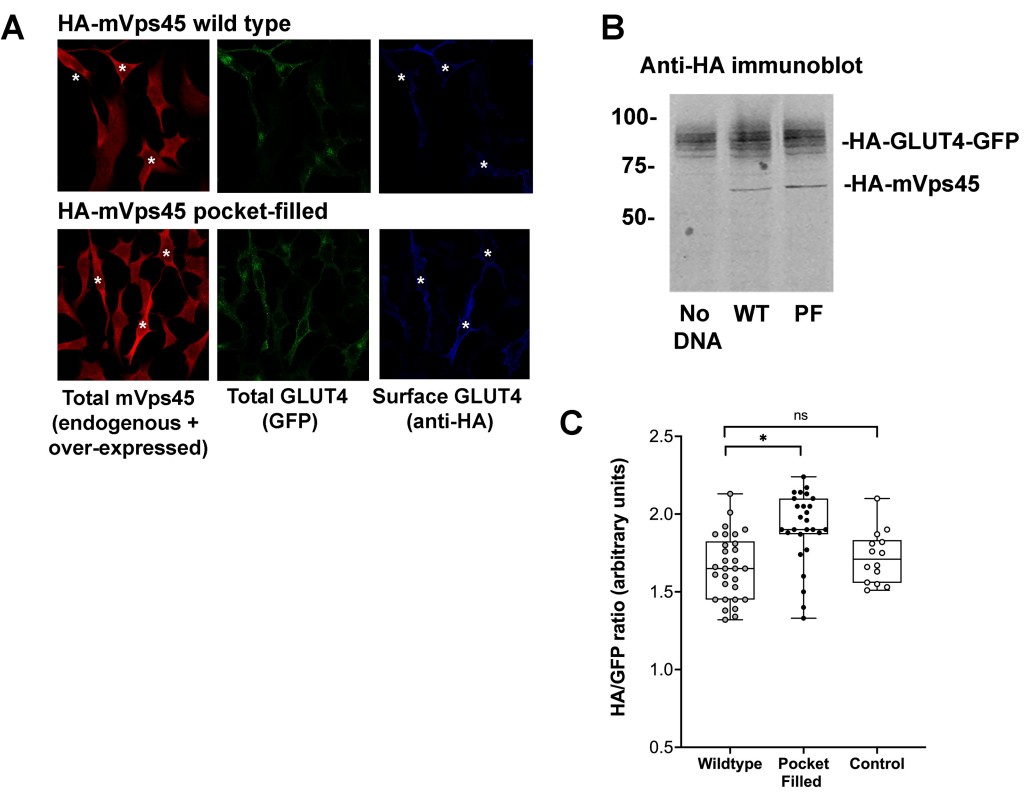

**Figure 2** **mVps45 binding to Sx16 controls GLUT4 trafficking.** HeLa cells expressing HA-GLUT4-GFP were transfected with plasmids encoding either HA-mVps45 (wild-type) or HA-mVps45-V107R (a mutant which prevents the interaction of the Sx16 N-terminus with mVps45 –see text). 48 h after transfection, cells were incubated in serum-free media for 2 h, fixed and cell surface GLUT4 immuno-stained using anti-HA (pseudo-coloured blue) prior to permeabilization. Subsequently, cells were permeabilised and stained using anti-mVps45 which detects both endogenous and over-expressed mVps45 (pseudo-coloured red; note that the use of HA-tagged mVps45 constructs and HA-tagged GLUT4 precluded this as a means to distinguish cells over-expressing mVps45 species). Signal from GFP is pseudo-coloured green. (A) Data from a typical experiment; white asterisk represent cells expressing higher than average anti-mVps45 immunoreactivity and which are therefore assumed to be over-expressing the indicated species. Data from a typical experiment is shown. (B) Levels of HA-GLUT4-GFP were not significantly altered upon over-expression of either wild-type or mVps45-V107R, which were expressed at similar levels (both are recognised by anti-HA); the approximate positions of molecular weight markers are shown (in kDa). (C) Quantification of the HA/GFP ratio from fields of cells such as those shown in (A). Fields of cells transfected with mVps45-V107R exhibited increased HA/GFP ratios compared to cells transfected with wild-type mVps45 (*$p < 0.001$ ANOVA). Wild-type transfected cells were indistinguishable from non-transfected controls (ns; $p = 0.33$ ANOVA). Each point on the graph is from a single field of cells; data from three independent biological experiments is presented.

the hypothesis that an interaction between the N-terminus of Sx16 and mVps45 is required for the correct trafficking of GLUT4.

## Sx16 is phosphorylated on T7

Previous work from this lab has shown that Sx16 is a phosphoprotein (*Perera et al., 2003*). We therefore sought to identify potential sites of phosphorylation of this key SNARE. Attempts to identify phosphosites using combinations of co-IP or purification from

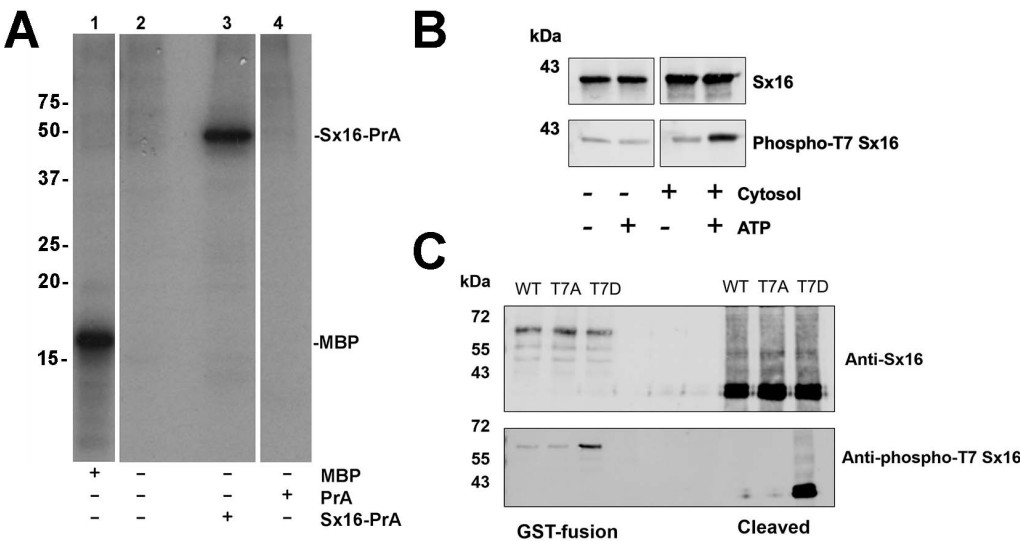

**Figure 3  Sx16 is phosphorylated on Threonine-7.** (A) 3T3-L1 adipocyte cytosol was used in a series of *in vitro* kinase reactions with [$^{32}$P] $\gamma$-ATP. Lane 1 shows that myelin basic protein (MBP) was efficiently phosphorylated by adipocyte cytosol. In lane 3, adipocyte cytosol phosphorylated recombinant Sx16-PrA, but did not phosphorylate PrA alone (lane 4). Lane 2 shows an identical reaction to that in lane 3 but Sx16-PrA was omitted. All lanes were run on the same gel, which has been cropped to remove either empty or irrelevant lanes; the data shown are all taken from the same exposure of the autoradiogram. This experiment was replicated with multiple batches of adipocyte cytosol with qualitatively similar results. (B) A similar experiment in which 0.5 μg of bacterially expressed Sx16 was incubated with or without 3T3-L1 adipocyte cytosol (30 μg; 'Cytosol') in the presence or absence of 25 μM ATP for 60 min at 30 °C as indicated. Samples were then immunoblotted with either anti-Sx16 antibodies ('Sx16' in figure) or anti-phospho-T7 antibodies ('Phospho-T7' Sx16 in figure). (C) A typical immunoblot in which equal amounts of wild-type Sx16, Sx16-T7A or Sx16-T7D were separated by SDS-PAGE and immunoblotted with anti-Sx16 or anti-phospho-T7 antibodies as indicated. This experiment shows data using intact GST-fusion proteins from bacterial lysates (left lanes; 20 μg /lane) or using purified Sx16 proteins released from Glutathione beads by thrombin ('Cleaved'; right lanes, 0.5 μg/lane). Data from a typical experiment is shown; the approximate positions of molecular weight markers are indicated (in kDa).

adipocytes proved unsuccessful. However, we consistently observed phosphorylation of recombinant Sx16 *in vitro* using an adipocyte cytosolic extract (Fig. 3A). Therefore, this phosphorylated recombinant Sx16 was subjected to phospho-site mapping. Phospho-mapping hits (FingerPrints ProteomicsFacility, University of Dundee) were assigned a score based upon probabilities that the fragment detected was a true phospho-peptide (Table 1; Fig. S1). The highest score, and the only hit, was assigned to fragments containing Threonine-7 (hereafter T7) suggesting that under these conditions, T7 is a major phospho-acceptor site. We therefore generated a phospho-selective antibody against the N-terminal 16 amino acids of murine Sx16. This antibody recognized a band of the expected molecular weight in *in vitro* phosphorylation reactions, the intensity of which increased significantly when ATP was included in the reactions (Fig. 3B). Consistent with this antibody recognising the phosphorylated forms of Sx16, we showed that the antibody cross-reacted strongly with recombinant Sx16-T7D but significantly less so with Sx16-T7A or Sx16 purified from bacteria (Fig. 3C).

**Table 1  Mascot data for phospho-site mapping identifies Sx16 Thr-7 as a site of *in vitro* phosphorylation .** Shown are the data confirming *in vitro* phosphorylation of Sx16 Thr-7. The data in *italics* are from a mis-cleaved peptide which also confirms the T7 site. Data was produced by the Fingerprints Facility at the University of Dundee.

| Spectra[*] | Observed | Mr (expt) | Mr (calc) | ppm | Miss | Score | Expect | Rank | Unique | Peptide |
|---|---|---|---|---|---|---|---|---|---|---|
| a | 571.3047 | 1,140.5948 | 1140.5944 | 0.37 | 0 | (62) | 5.8e−07 | 1 | U | R.LTDAFLLLR.N + Phospho (ST) |
| b | 571.3047 | 1,140.5948 | 1140.5944 | 0.37 | 0 | (55) | 2.8e−06 | 1 | U | R.LTDAFLLLR.N + Phospho (ST) |
| c | *433.2391* | *1,296.6955* | *1,296.6955* | *−0.00* | *1* | *(48)* | *1.6e−05* | *1* | *U* | *R.R LTDAFLLLR.N + Phospho (ST)* |

**Notes.**
  [*]The corresponding spectra are shown in Fig. S1.

## Phospho-mimetic mutation of T7 perturbs binding to mVps45

Interaction of Sx16 with mVps45 involves the extreme N-terminus of Sx16 (*Carpp et al., 2006*; *Dulubova et al., 2002*; *Furgason et al., 2009*; *Munson & Bryant, 2009*), hence the identification of a potential phospho-acceptor site in this region could suggest a potential regulatory mechanism to control Sx16/mVps45 interaction and function. To test this, we employed recombinant Sx16-PrA to pull-down HA-tagged human mVps45 expressed in yeast and compared wildtype Sx16 (which is not phosphorylated when expressed in bacteria) with Sx16-PrA containing the phospho-mimetic T7D mutation (Fig. 4). As shown wild-type Sx16 effectively pulled-down HA-tagged mVps45 from yeast lysate. Although Sx16-T7D was still able to bind HA-mVps45, this was reduced in both magnitude and affinity (Fig. 4). This data strongly suggests that phosphorylation of T7 could significantly impact Sx16 function by preventing or reducing the affinity of its interaction with mVps45.

## Sx16-T7D perturbs insulin-stimulated glucose transport

Previously, we have shown that perturbation of either Sx16 or mVps45 function modulated GLUT4 sorting in adipocytes. Hence, if the interaction between Sx16 and mVps45 is regulated by phosphorylation at T7, as suggested by the data in Figs. 3 and 4, we would predict that phosphorylation of this site would be expected to modulate GLUT4 trafficking *in vivo*.

To test this hypothesis, full length Sx16 wild-type, or T7A and T7D mutants were over-expressed in 3T3-L1 adipocytes, a well-used model of insulin-stimulated GLUT4 translocation and glucose transport. We consistently observed reduced insulin-stimulated 2-deoxyglucose uptake in cells over-expressing Sx16-T7D compared to either wild-type or Sx16-T7A (Fig. 5A). To facilitate comparison of datasets from biological replicates in which levels of over-expression of Sx16 varied and the magnitude of the insulin response varied, the data from multiple experiments expressed as a % of the maximal insulin response is shown in Fig. 5A. A representative immunoblot of lysates from a typical experiment is shown in Fig. 5B; for reasons that we have been unable to identify, the overexpressed proteins migrate faster than endogenous Sx16. The apparent reduction in total GLUT4 levels in T7D-expressing cells in this dataset was not recapitulated consistently in all experiments. Importantly, insulin-stimulated activation of Akt was not modulated by expression of the different Sx16 mutants, indicating that a deficit in insulin-signalling is unlikely to be the mechanism responsible for reduced insulin-simulated glucose transport (Fig. 5C).

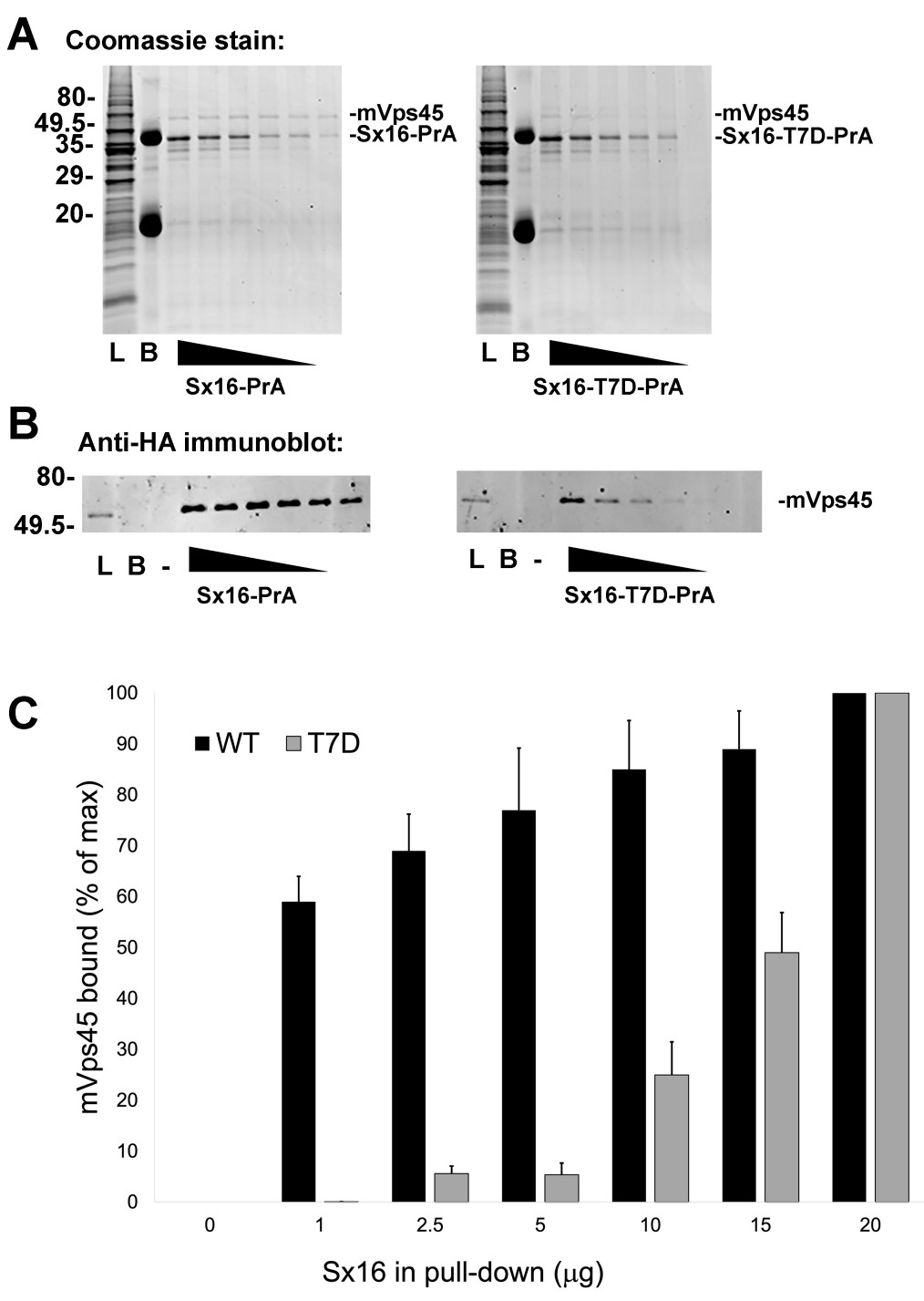

**Figure 4   A phospho-mimetic mutation of Sx16-T7 exhibits reduced interaction with mVps45.**
Identical amounts of recombinant Protein A-tagged Sx16 wild type or Sx16-T7D were used to pull down
HA-tagged mVps45 over-expressed in yeast. A constant amount (250 μg) of yeast lysate expressing
HA-mVps45 was incubated with decreasing amounts (20, 15, 10, 5, 2.5, and 1 μg) of either Sx16-PrA or
Sx16-T7D-PrA for 2 h on ice. PrA-tagged species were then captured using IgG-Sepharose beads, washed
and eluted by boiling in SDS-PAGE sample buffer. (continued on next page...)

**Figure 4 (…continued)**
Samples were then stained for total protein (A) or immunoblotted using anti-HA (B). In the figure, **L** refers to starting Lysate and B shows a sample of IgG beads ( 10x more than used in the capture lanes) to control for any non-specific binding. PrA alone did not exhibit any binding to HA-mVps45 (uncropped blots can be found in the Supplemental Information). Data from a representative binding experiment is shown, together with quantification of three separate experiments of this type presented in (C) as mean ± S.D. Note that in the quantification, the data are normalised relative to the level of mVps45 binding observed using 20 μg of Sx16. The approximate positions of molecular weight markers are indicated (in kDa).

## Sx16 is a substrate for AMP-dependent protein kinase

The observation that Sx16-T7 is a potential regulatory site for the control of GLUT4 trafficking prompted an investigation of the kinase(s) that might phosphorylate this site. The peptide sequence around T7 resembles the proposed consensus sequence for the AMP activated protein kinase (AMPK) and several of its related kinases (*Bright, Thornton & Carling, 2009*). AMPK activation enhances insulin-stimulated glucose transport in skeletal muscle (*Kleinert et al., 2017*). However, it should be noted that in 3T3-L1 adipocytes, the converse is true: activation of AMPK inhibits insulin-stimulated glucose transport and GLUT4 translocation to the plasma membrane (*Salt, Connell & Gould, 2000*). Using recombinant Sx16 as a substrate for *in vitro* phosphorylation reactions revealed that AMPK phosphorylated Sx16 *in vitro*, and that the extent of this phosphorylation was comparatively reduced in Sx16-T7A under identical initial rate phosphorylation conditions. Importantly phosphorylation was enhanced by inclusion of AMP in the kinase assay mix (Fig. 6). Sx16 was also phosphorylated by the AMPK-related kinase SIK2 (*Bright, Thornton & Carling, 2009*; *Patel et al., 2014*) and the extent of this phosphorylation was consistently reduced in the Sx16-T7A mutant (Fig. 6); MARK1 (*Bright, Thornton & Carling, 2009*) also phosphorylates Sx16 but this was not reduced in Sx16-T7A and hence was not studied further (Fig. S2). By contrast the AMPK-related kinase BRSK2 (*Bright, Thornton & Carling, 2009*) (Fig. S2) was without effect on Sx16 phosphorylation. It should be noted that phosphorylation of Sx16-T7A by AMPK is also observed, indicating that further phospho-acceptor sites are present within this SNARE protein.

Our initial report identifying Sx16 as a phosphoprotein indicated that total Sx16 phosphorylation levels were slightly reduced in insulin-stimulated 3T3-L1 adipocytes compared to controls (*Perera et al., 2003*). Hence, we adopted several routes to address this in intact 3T3-L1 adipocytes here focussing on phospho-T7 levels using our phospho-selective antibody. We have been unable to measure any change in phospho-T7 levels in 3T3-L1 adipocytes in response to insulin (Fig. 7A) or 5-aminoimidazole-4-carboxamide ribonucleotide (AICAR) a known activator of AMPK, (Fig. 7B). One possible explanation for this is provided by the data in Fig. 7C in which lysates from Mouse Embryonic Fibroblasts (MEFs) engineered to lack AMPK compared to their wildtype counterparts were used in *in vitro* phosphorylation reactions with or without the addition of ATP. Lack of AMPK modestly reduced the phosphorylation of Sx16 (Fig. 7D), but the effect was small, indicating that other, presently unidentified, kinases may regulate this site. This is consistent with the autoradiography in Fig. 6 which indicates substantial phosphorylation of Sx16-T7A by AMPK, suggesting that other sites within Sx16 may be important, and with our previous
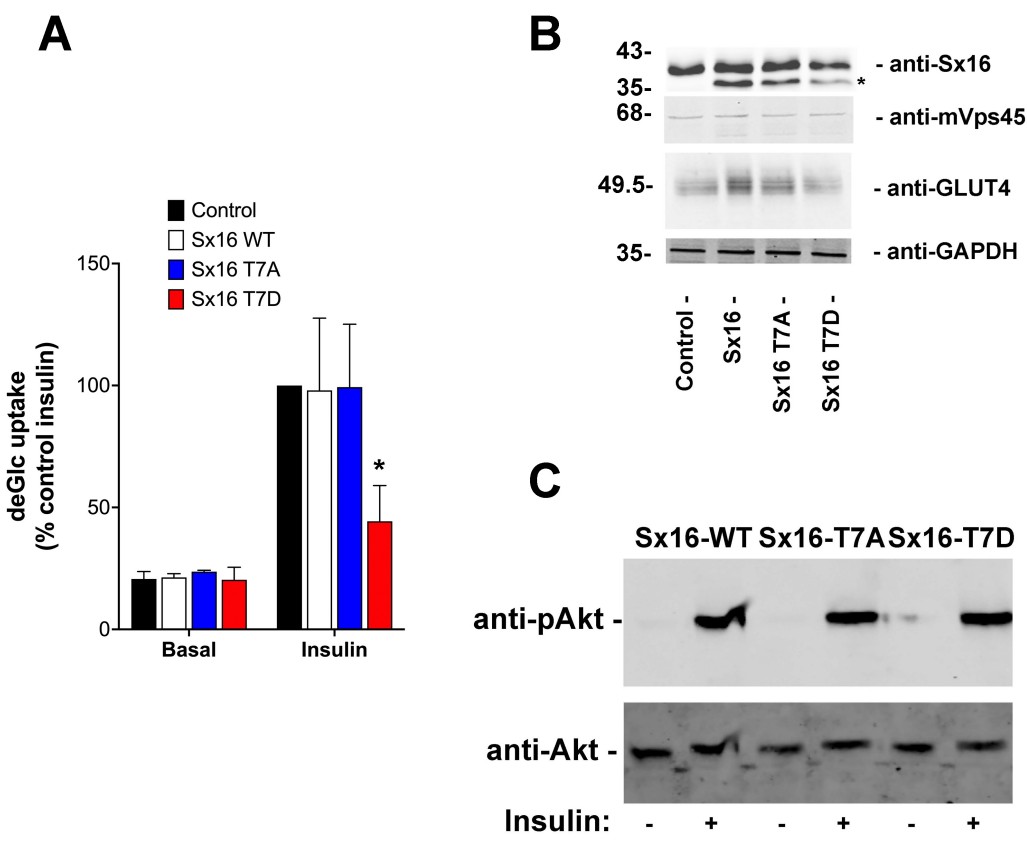

**Figure 5  Sx16-T7D reduces insulin-stimulated glucose transport.** 3T3-L1 adipocytes were infected with lentivirus delivering either wild-type Sx16, or Sx16-T7A or Sx16-T7D as outlined in *Methods*. (A) DeGlc uptake was assayed after incubation with or without 1 μM insulin for 30 min. Shown are the means of three independent experiments in which basal and insulin-stimulated deGlc uptake rates were measured in quadruplicate at each condition and are presented as a % of the insulin-stimulated rate in control (non-infected) cells. Over-expression of Sx16-T7D consistently impaired insulin-stimulated deGlc uptake; $p = 0.02$ compared to control cells (ANOVA). No other differences were observed between groups. (B) Shown are lysates from a typical dataset immunoblotted with anti-Sx16, anti-GLUT4, anti-mVps45 and anti-GAPDH. Note that over-expressed Sx16 and mutants thereof consistently migrate faster than the endogenous protein (indicated by * on the figure), the approximate positions of molecular weight markers are indicated (in kDa). We saw no significant differences in levels of expression of the different Sx16 mutants across all experiments of this type. (C) 3T3-L1 adipocytes expressing either Sx16-WT, T7A or T7D (as indicated) were treated with or without 1 μM insulin for 30 min and lysates separated on SDS-PAGE and immunoblotted using antibodies that recognise phosphorylated Akt or total Akt, as shown. Data from a typical experiment is shown.

data (*Perera et al., 2003*). Future work should address whether phosphorylation status of Sx16 is altered in disease or conditions of insulin resistance.

## DISCUSSION

Previous work from our group and others has established a role for Sx16 in the trafficking of GLUT4 (*Bremner et al., 2022*; *Proctor et al., 2006*; *Shewan et al., 2003*). Similarly, we have shown that mVps45, the Sec1/Munc18 protein that is known to bind to Sx16 is also

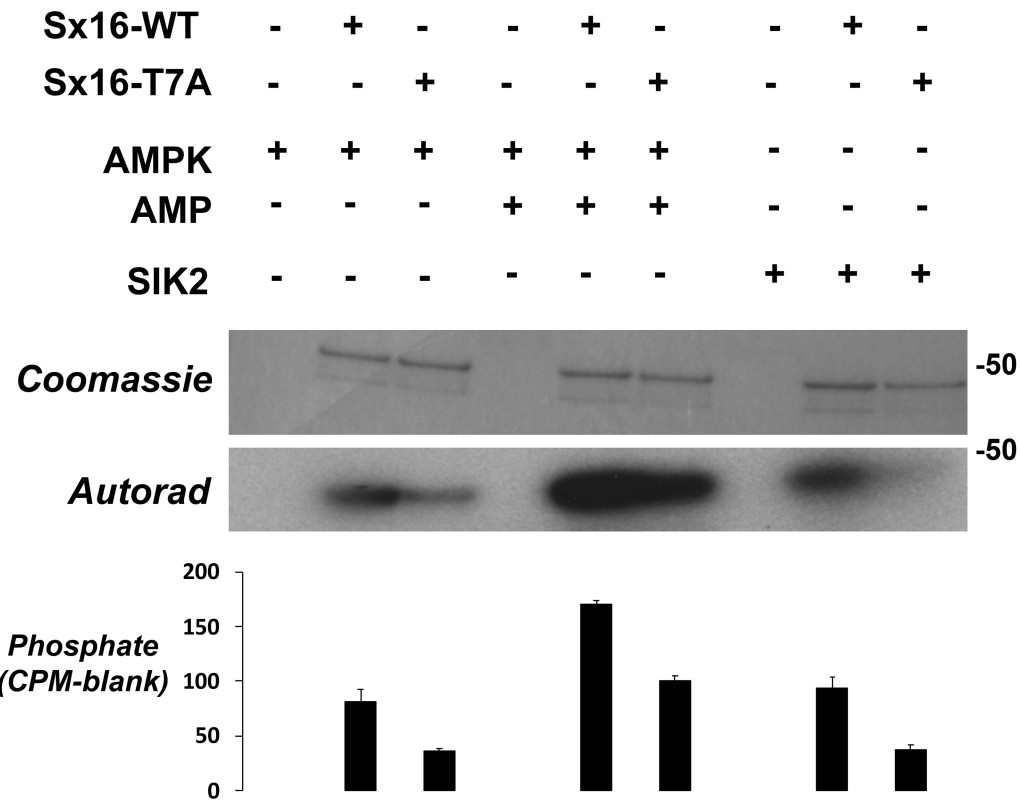

**Figure 6** **Sx16 is phosphorylated by AMPK *in vitro* on T7.** Purified Sx16 (WT or T7A mutant) was incubated with 5mU of either AMPK or SIK2 as indicated plus radiolabelled Mg.ATP for 30 mins at 30 °C as described. The reaction was stopped by addition of 4x SDS buffer and the protein visualised following SDS-PAGE by Coomassie Staining. Phosphate incorporated into Sx16 was detected by autoradiography and quantified by excising the Sx16 bands and Cerenkov counting. A representative protein stain and autoradiogram are shown, while average (±s.e.m.) CPM incorporated in two separate experiments is provided in the bar chart. The approximate positions of molecular weight markers are indicated (in kDa).

required for this trafficking event (*Roccisana et al., 2013*). Studies from our group and others has shown that Sx16 binds to mVps45 *via* the extreme N-terminus of Sx16 inserting into a pocket on the surface of mVps45 (*Carpp et al., 2006*; *Carpp et al., 2007*; *Munson & Bryant, 2009*; *Struthers et al., 2009*). Disruption of either the Sx16 N-terminus or the V107R mVps45 mutant prevents this interaction (*Carpp et al., 2006*; *Dulubova et al., 2002*; *Shanks et al., 2012*; *Stepensky et al., 2013*). Here we show that perturbation of interaction between these two proteins using a peptide mimetic corresponding to the N-terminal 33 residues of Sx16 reduces insulin-stimulated GLUT4 translocation (Fig. 1). Further, over-expression of the V107R mutant of mVps45 (V107R; Fig. 2) also perturbs GLUT4 trafficking in HeLa cells, consistent with an important role for the interaction of these proteins in determining the intracellular fate of GLUT4. It is important to emphasise that these two experiments are distinct: overexpression of the Sx16 N-terminal peptide competes with endogenous Sx16 for binding to mVps45 and thus inhibits *productive* SNARE complex formation (*Furgason et al., 2009*; *Munson & Bryant, 2009*; *Struthers et al., 2009*). The mutant form

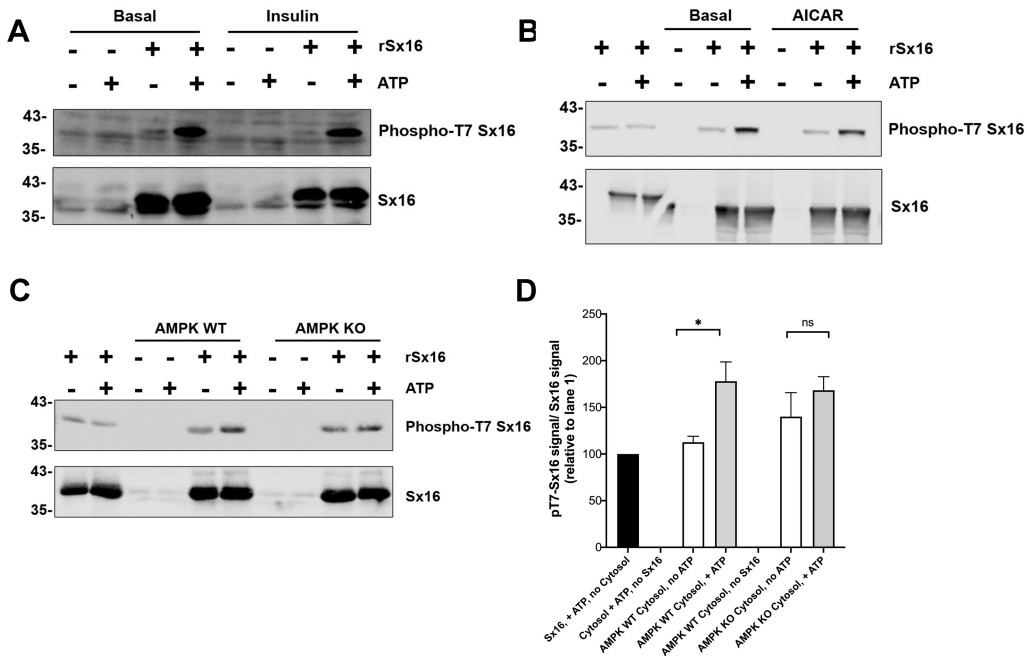

**Figure 7  Cytosol from insulin- and AICAR-treated cells was without effect on Sx16 phosphorylation.**
(A) 0.5 μg of bacterially expressed Sx16 was incubated with 30 μg cytosol prepared from 3T3-L1
adipocytes treated without (Basal) or with 100 nM insulin as shown in the presence or absence of 25
μM ATP for 60 min at 30 °C as indicated. Samples were then immunoblotted with anti-Sx16 antibodies
('Sx16' in figure) or anti-phospho-T7 antibodies ('Phospho-T7 Sx16' in figure). (B) similar experiment to
(A) except that cytosol from basal (untreated) cells was compared to cytosol prepared from cells treated
with 1 mM AICAR for 60 min as indicated. As a control, cytosol was omitted from the left two lanes of
this experiment. In (C) Sx16 was incubated with cytosol from wild type MEFs or from MEFs lacking
AMPK α1 and α2 in the presence or absence of ATP as indicated. As a control, cytosol was omitted from
the left two lanes of this experiment. (D) Quantification of three experiments of this type. * indicates a
statistically significant difference, $p = 0.0064$ comparing signal ATP in the presence of WT MEF cytosol
by unpaired $t$-test. The similar comparison for AMPK KO cytosol was not significantly different; $n = 3$
biological replicates. In all figures the approximate positions of molecular weight markers are indicated to
the left of the panels (in kDa).

of mVps45-V107R (Fig. 2) cannot bind the Sx16 N-terminus but can still bind Sx16 in
a conformation incompatible with productive SNARE complex formation (*Furgason et
al., 2009*; *Munson & Bryant, 2009*; *Struthers et al., 2009*). As mVps45 cycles on and off
membranes and through these different conformational states (*Bryant & James, 2003*), the
mechanism underlying the inhibitory effects observed in Figs. 1 and 2 is likely distinct.
However, these data clearly establish an important role for the interaction of the N-terminus
of Sx16 with mVps45 in the sorting of GLUT4 to IRVs.

This begs the question of how this interaction might be regulated. Recent work has
identified phosphorylation as a potential control mechanism for the function of Q
$_a$-SNARE/Syntaxin family members (*Aran, Bryant & Gould, 2011*; *Barak-Broner et al.,
2021*; *Chung, Polgar & Reed, 2000*; *Gurunathan et al., 2002*; *Liu, Heidelberger & Janz, 2014*;
*Marash & Gerst, 2001*; *Marash & Gerst, 2003*; *Tadokoro et al., 2016*). Sx3 is constitutively
phosphorylated at T14 in mast cells, and this phosphorylation has been shown to regulate

the interaction of this $Q_a$-SNARE with its cognate Sec1/Munc18 protein, Munc18-2 (*Tadokoro et al., 2016*). Sx1 phosphorylation at S14 has also been shown to be a key regulatory event in controlling exocytosis in neuroendocrine cells (*Barak-Broner et al., 2021*). It is of note that both these proteins are phosphorylated in their N-terminal regulatory domains (*MacDonald, Munson & Bryant, 2010*; *Munson & Bryant, 2009*; *Sudhof & Rothman, 2009*).

We (*Perera et al., 2003*) and others (*Fazakerley et al., 2022*; *Hornbeck et al., 2004*) identified Sx16 as a phosphoprotein; using recombinant Sx16 and concentrated adipocyte cytosol, we identified T7 as a potential phospho- acceptor site (Fig. 3, Table 1, Fig. S1). This is intriguing, as this site lies within the domain of Sx16 known to bind mVps45 (*Carpp et al., 2006*; *Dulubova et al., 2002*; *Furgason et al., 2009*), and others have suggested that phosphorylation of Sx proteins in this region might modulate SM protein binding (*Barak-Broner et al., 2021*; *Munson & Bryant, 2009*; *Tadokoro et al., 2016*). We therefore generated recombinant phospho-mimetic Sx16-T7D and its wild-type counterpart in bacterial and compared their ability to pull-down mammalian mVps45 expressed in yeast. Figure 4 shows that the phospho-mimetic mutant, Sx16-T7D, was significantly impaired in its ability to pull-down mVps45, suggesting that phosphorylation of this site might regulate Sx16/mVps45 interaction and thus impair trafficking events dependent on the productive Sx16/mVs45 interaction. Consistent with this result, studies of the yeast $Q_a$-SNARE Sso2p have shown that dephosphorylation of this $Q_a$-SNARE drives the formation of SNARE complex assembly (*Gurunathan et al., 2002*; *Marash & Gerst, 2001*; *Marash & Gerst, 2003*). This supports our working model in which cycles of phosphorylation/dephosphorylation of Sx16 modulate the interaction of Sx16 with mVps45 and thus control trafficking events mediated by these proteins.

To further test this hypothesis, we found that over-expression of Sx16-T7D, but not T7A (or wild-type Sx16) impaired insulin stimulated glucose transport in 3T3-L1 adipocytes (Fig. 5). Although the magnitude of this inhibition exhibited some variation reflecting different levels of over-expression relative to endogenous Sx16, the T7D mutant consistently impaired insulin-stimulated glucose transport ($55 \pm 16\%$). This data is consistent with a model in which phosphorylation of the N-terminus of Sx16 prevents the formation of productive SNARE complexes and thus effective sorting of GLUT4 into IRVs. It is important to note that although the affinity of Sx16-T7D for mVps45 is reduced compared to non-phosphorylated Sx16 (Fig. 4), the net effect is to reduce the formation of productive SNARE complexes (see 'Discussion' above) and thus perturb trafficking of GLUT4 into IRVs (Fig. 5) (*Bremner et al., 2022*; *Proctor et al., 2006*; *Roccisana et al., 2013*).

This data suggests that phosphorylation of T7 might play a role in the regulation of glucose disposal and thus we sought to identify which kinase(s) may be potential regulators of this site. The peptide sequence around T7 resembles the proposed consensus sequence for the AMP activated protein kinase (AMPK) and several of its related kinases (*Bright, Thornton & Carling, 2009*). Using *in vitro* kinase assays, we show that both AMPK and SIK2 phosphorylate Sx16 *in vitro* (Fig. 6); however it is important to note that the data in Fig. 6 reveal that Sx16 is probably phosphorylated on multiple sites by AMPK *in vitro*, as Sx16-T7A is phosphorylated by AMPK, albeit to a lesser degree than wildtype Sx16. It

should be noted that other AMPK-family members (SIK1, MARK2, NUAK and BRSK2) did not phosphorylate this or indeed other sites in Sx16 (Fig. 6).

Given the importance of T7 in regulating interaction with mVps45, we used our phospho-specific antibody to ascertain whether there were insulin- or AICAR-driven changes in Sx16 phosphorylation at this site (Figs. 7A, 7B). No significant changes in T7 phosphorylation were observed using cytosol from insulin or AICAR-treated adipocytes. We have previously reported a decrease in Sx16 phosphorylation in response to insulin using immunoprecipitation from adipocytes. It is possible that elements of the regulatory network are not recapitulated using cytosolic extracts of cells, such as those shown in Fig. 7. It is frustrating that our phospho-specific antibody proved insufficiently discriminatory to study phosphorylation of endogenous Sx16 in whole cell extracts (by way of example, see Fig. 7A in lanes lacking recombinant Sx16). Hence, it is formally possible that the experimental results reported in Fig. 7 do not fully reflect the situation in intact cells. Furthermore, it is possible that the insulin-regulated dephosphorylation of Sx16 observed in Perera et al. is accounted for by changes at sites other than T7.

In an effort to ascertain whether AMPK was the predominant kinase for T7, we prepared lysates from MEFs engineered to lack AMPK ('AMPK KO cells') and found that phosphorylation of Sx16 was modestly reduced when using MEFs from AMPK KO cells compared to their wildtype counterparts. While supporting the hypothesis that AMPK phosphorylates Sx16, clearly other potential explanations for this result should be kept in mind, such as inadequate activation of a distinct down-stream AMPK, *etc*.

These caveats notwithstanding, our data raise the interesting possibility that activation of AMPK (a known regulator of glucose transport (*Bright, Thornton & Carling, 2009*; *Salt, Connell & Gould, 2000*)) or SIK2 could impact GLUT4 sorting by targeting the Sx16/mVps45 interaction. While we did not observe any consistent effect of either insulin or AICAR on Sx16 phosphorylation at this site, our data indicates that other, presently unidentified kinases, may target this site and thus further work is required to functionally link phosphorylation of T7 to specific cellular stimuli or diseased states.

In sum, our data confirm an important role of Sx16/mVps45 in GLUT4 sorting in adipocytes and reveal that phosphorylation of Sx16 might offer a novel target for therapeutic intervention in Type-2 diabetes.

# ACKNOWLEDGEMENTS

We thank Drs Douglas Lamont and Kenneth Beattie of the FingerPrints Proteomics Facility, Dundee, for help with the MS analysis and Mike Mao at Antagene for help with the phospho-antibody production. We thank the reviewers of this work for constructive and helpful suggestions.

## Funding

This work was supported by Diabetes UK grants 13/0004725 to NJB and GWG and 18/0005847 to Gwyn W. Gould, Nia J. Bryant and Calum Sutherland, a Colin MacArthur

PhD Studentship award to JR and British Heart Foundation grant PG/12/3/29344 to Calum Sutherland. The funders had no role in study design, data collection and analysis, decision to publish, or preparation of the manuscript.

### Grant Disclosures

The following grant information was disclosed by the authors:
Diabetes UK: 13/0004725, 18/0005847.
Colin MacArthur PhD Studentship.
British Heart Foundation: PG/12/3/29344.

### Competing Interests

Gwyn W Gould is an Academic Editor for PeerJ.

### Author Contributions

- Shaun K. Bremner conceived and designed the experiments, performed the experiments, analyzed the data, prepared figures and/or tables, authored or reviewed drafts of the article, and approved the final draft.
- Rebecca Berends conceived and designed the experiments, performed the experiments, analyzed the data, prepared figures and/or tables, authored or reviewed drafts of the article, and approved the final draft.
- Alexandra Kaupisch conceived and designed the experiments, performed the experiments, analyzed the data, prepared figures and/or tables, authored or reviewed drafts of the article, and approved the final draft.
- Jennifer Roccisana conceived and designed the experiments, performed the experiments, analyzed the data, prepared figures and/or tables, authored or reviewed drafts of the article, and approved the final draft.
- Calum Sutherland conceived and designed the experiments, performed the experiments, analyzed the data, prepared figures and/or tables, authored or reviewed drafts of the article, and approved the final draft.
- Nia J. Bryant conceived and designed the experiments, analyzed the data, prepared figures and/or tables, authored or reviewed drafts of the article, and approved the final draft.
- Gwyn W. Gould conceived and designed the experiments, analyzed the data, prepared figures and/or tables, authored or reviewed drafts of the article, and approved the final draft.

### Data Availability

The raw data and immunoblots are available in the Supplemental Files.

### Supplemental Information

Supplemental information for this article can be found online at http://dx.doi.org/10.7717/peerj.15630#supplemental-information.

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
