# Peer review of "Phosphorylation of the N-terminus of Syntaxin-16 controls interaction with mVps45 and GLUT4 trafficking in adipocytes"

_PeerJ, doi:10.7717/peerj.15630_

## Round 0.1 · original submission · Major Revisions

Your article has been reviewed by three experts in the field that raised various concerns, many of which may be addressed by changes/discussions in the text. Please make sure to address the question as to why the T7D mutation results in a dominant-negative phenotype.

Further, some comments should be addressed experimentally as outlined by reviewer #1 including but not necessarily limited to the question if GLUT4 translocation is impaired (Fig 5), if the T7D mutation destabilizes the protein, and if there is a different splice-isoform in 3T3-L1 cells.

Reviewer 1 ·

Basic reporting

No comment.

Experimental design

No comment.

Validity of the findings

The manuscript by Bremner et al. characterizes the phosphorylation of the SNARE protein, syntaxin-16 (Sx16), and studies how this phosphorylation influences the interaction of Sx16 with the Sec1/Munc18-like protein Vps45 and how it influences GLUT4 glucose transporter translocation. This is an important area for understanding how insulin stimulates glucose uptake, and the work contributes new data in this area. In general, the studies are well done, however there are some points that require additional data to be fully convincing.

The main deficiency is in Figure 5, which implies that expression of Sx16-T7D (containing a putative phosphor-mimetic mutation) in 3T3-L1 adipocytes inhibits the ability of insulin to stimulate glucose uptake. In the data shown, the T7D mutant is expressed at substantially lower abundance than the endogenous Sx16 protein. Since the effect of this protein is presumably from a competitive mechanism, whereby it out-competes the endogenous protein for some of its binding partners, it is not clear how the exogenous mutant protein can have this effect. Additionally, only glucose uptake data are show, and there is no indication of whether GLUT4 translocation is also impaired. Insulin signaling is not examined; presumably this is intact but it is an important control that should be shown. Finally, the exogenously expressed wildtype and mutant proteins run faster than the endogenous Sx16 proteins on SDS-PAGE. The authors are not able to account for this observation, but they have not examined whether the main splice form that is expressed in 3T3-L1 cells is different from the one they are expressing exogenously.

Other points are below:

The text states that the V107R mutation is on the outer surface of Vps45, but the same mutation is referred to as a “pocket filled” mutant protein. If the mutation fills a binding pocket for the Sx16 N-peptide then is it really on the outer surface? This is confusing.

Fig. 2 presents data from transfected HeLa cells, similar to Fig. 1, but it is not clear if the cells are basal or insulin-stimulated. If they were not serum starved, then it should be so stated. Would this be more analogous to the basal or insulin stimulated state, or to something in between?

Decreased binding affinity in Fig. 4 based on T7D mutation. Can a phosphorylated peptide be used? This might have an even larger effect, since only the charge and not the size of the phosphate will be mimicked by the T7D mutation.

In Fig. 5, the abundance of exogenous Sx16 T7D seems to be decreased, relative to exogenous WT and T7A proteins. Does the T7D mutation destabilize the protein, or is this just due to differences in infection of the cells with viral constructs? GLUT4 abundance is also reduced, although it is noted in the text that the reduced abundance of GLUT4 was not observed consistently. More concerning, as noted above, the exogenous Sx16 proteins must out-compete the endogenous protein, it is uncertain how the exogenous T7D mutant can have an effect. This protein appears to be expressed at much lower abundance compared to the endogenous, wildtype protein, so how could it displace binding partners of the endogenous protein to affect GLUT4 trafficking and glucose uptake?

The authors are not sure why the exogenous Sx16 runs faster than the endogenous protein on SDS-PAGE. Is this due to different splice forms, Stx16A and Stx16B? Stx16A has an additional 21 residues, which could account for the different apparent mass.

Fig. 5 would be more convincing if there were some data to support the idea that GLUT4 translocation is disrupted in 3T3-L1 adipocytes. For example, subcellular fractionation, or cell surface biotinylation, or use of a tagged reporter could be used, and would help to support the glucose uptake data.

The data in Fig. 7 were generated by incubating recombinant proteins with cytosol. Can the endogenous phosphorylated Sx16 protein be detected? If this is difficult in whole cell lysates, then perhaps total membranes could be used (either by pelleting a homogenate, or using a sodium carbonate preparation). Perhaps the authors could state what they have done toward this end. As a related question, can it be estimated what fraction of Sx16 is phosphorylated in 3T3-L1 cells? The phospho-T7 antibody looks reasonably specific on western blots. Can it be used to immunoprecipitate the phosphorylated protein? Does immunofluorescence using this antibody identify a particular location where Sx16 is phosphorylated? It is understood that these questions may be beyond the scope of the article the authors wish to publish in PeerJ, yet if it is possible to add some of these data it would be helpful to clarify the overall picture.

Additional comments

No comment.

·

Basic reporting

The article is written clearly and in professional English. There are a couple of typographical errors I spotted and a few suggestions for avoiding ambiguity in the additional comments.

Experimental design

The experimental design and execution is generally good. Research question is well defined.
As mentioned in additional comments I am not so sure of the rationale behind the design of the experiment for results shown in figure 2. Intuitively I do not see how expression of the pocket filled 'reduced binding' mutant interferes, as the endogenous protein which should still be able to bind and direct trafficking is still present, unless I have misunderstood something. Maybe the rationale can be more clearly stated.

Validity of the findings

All data has been provided. The reduced phosphorylation in AMPK KO experiment does not seem to be statistically significant. In the results in particular I feel the narrative describing this result are a bit misleading. See additional comments.

Additional comments

The paper is interesting and generally very well written.
Figure 1 supports the conclusion that in cells that express a Sx16 N-terminal peptide that competes for binding to Vps45 with endogenous Sx16 that Glut4 insulin stimulated translocation is reduced.
The narrative describing figure 2 could be a bit clearer in my opinion. The switching between V107R and ‘pocket filled’ nomenclature was a bit inconsistent. I would choose one or the other nomenclature after the first mention in the figure and narrative. I feel it needs to be made absolutely clear that the HA/GFP ratio is surface HA/GFP thus only measuring HA associated with Glut4 at the cell surface. I do not quite understand how expression of a mutant Vps45 that has reduced binding to Sx16 will interfere with binding of endogenous Vps45 to Sx16.
The results from figure 1 and 2 are different. In Fig1 the Glut4 present at the cell surface is unaltered in basal conditions when Syntaxin16/Vps45 interaction is likely disrupted. In figure 2 the Glut 4 present at the cell surface under basal conditions is increased when V107R mutant (which does not bind well to Sx16) is expressed. For Figure 2 the insulin stimulated scenario is not shown. I feel that the discussion needs to be expanded slightly to reflect these differences and perhaps speculate mechanistically on how Glut4 trafficking is being perturbed differently in the two experiments.
Figure 3 demonstrates quite convincingly that recombinant Sx16 can be phosphorylated in vitro by something in an adipocyte cytosol and that a phospho-specific antibody can distinguish between phosphorylated and non-phosphorylated syntaxin16
Figure 4 Does support the claim that the phosphomimetic mutant of Syntaxin16 has reduced binding to Vps45
Figure 5 shows that expression of the phosphomimetic mutant with reduced binding to Vps45 has a reduction in insulin stimulated Glut4 transport to the cell surface. This is consistent with the result in figure 1 which suggests issues with sorting Glut4 to an insulin sensitive compartment when the Sx16 Vps45 interaction is perturbed. Makes me further question the mechanism behind results shown in figure 2
Figure 6 identifies candidate kinases (AMPK and possibly SIK2responsible for the phosphorylation of Sx16
The authors attempt to address the phosphorylation status of Sx16 under different conditions but do not see a difference in phosphorylation in response to insulin or AICAR. In the narrative the authors state that lack of AMPK ‘significantly’ reduced the phosphorylation of Sx16 (Fig 7D). I do not feel this statement is supported statistically. This is also mentioned in the discussion which should be removed in my opinion.
Discussion is clearly written. A couple of suggestions are made above for inclusion/exclusion from the discussion

Minor typographical errors and suggestions
Abstract: The ability of insulin to stimulated…… (to stimulate).
In final sentence I would probably make it clear that T7 is in Sx16

Reviewer 3 ·

Basic reporting

This is a generally well-written, clear paper, demonstrating a potentially important mechanism for the regulation of insulin-stimulated glucose uptake in adipocytes. The authors provide good data and appropriate interpretation. There are some minor points that need addressing:

The summary of results at the end of the introduction seemed unusually and unnecessarily long (also repetitive with results and discussion). It does rather pre-empt the results section itself. Please shorten and simplify this and just keep a very brief summary of the main conclusions.

Please define abbreviations in the main text on first use: Sx16 (line 74), AICAR (line 317), MEFs (line 319).


Minor typographical errors, figure legends:
Fig. 1: “asterixis” (spelling)
Fig. 2: “asterixis” and “anti anti-mVps45”
Fig. 3: “Sc16-T7D”
Fig. 3B: the labels under the lanes are not aligned.
Fig. 5: “N0”
Fig. 7B: Is the apparently higher molecular weight of Sx16 in the left lanes due to the running of the gel?

Please add “Sx16” to the title for Table1, for clarity (“Thr 7 of Sx16…”)

Experimental design

Reference and explanation for AICAR addition needs to be included in the introduction, as it is unclear why this is added in Fig. 7 (or indeed why this would be significant).

What was the source of the “QuickChange Kit” (line 128)?
No method is given for immunofluorescence with permeabilised or intact cells (Fig. 1 and 2), and this should be added.

Validity of the findings

The data are generally well presented and support the claims in the paper. However, whilst the data shown does make the point, the statistics used are poorly explained, and often not mentioned. For example, from the legend to Fig. 1: “ *indicates a significant reduction in insulin-stimulated surface/total ratio compared to control and F10A insulin-stimulated cells (p=0.02)”. What test was used here to determine significance? Similar comments apply to other figures too, and this information should be added.

Additional comments

This report shows that syntaxin 16 (Sx16) is phosphorylated at the N-terminus, residue 7. The identification of the phosphorylated residue is a novel finding, as is the demonstration that this acts as a potential regulatory mechanism in the binding of Sx16 to mVps45. Importantly, the authors show that phosphorylation of Sx16 may reduce insulin-stimulated glucose transport. The kinase that acts on Sx16 is unclear, but may be AMPK or SIK2.

---

## Round 0.2 · accepted · Accept

Your article was re-reviewed by the original three reviewers and I am delighted to inform you that all reviewers thought you satisfied their comments. Congratulations!

Reviewer 1 ·

Basic reporting

No comment.

Experimental design

No comment

Validity of the findings

No comment.

Additional comments

The authors have done a nice job of responding to previous critiques. The manuscript is much improved and in my opinion it is now acceptable for publication.

·

Basic reporting

The authors have addressed all the points I made in the initial review. I have no further comments

Experimental design

N/A

Validity of the findings

N/A

Additional comments

N/A

Reviewer 3 ·

Basic reporting

The manuscript has been amended according to the comments of three reviewers. It is now clearer, with some additional data added, and I believe all concerns have been addressed.

Experimental design

This is now clearer, with missing methods now added and more explanation of certain points.

Validity of the findings

Reference to the statistical methods used has now been added.

Additional comments

I have no further comments.